# Computational Prediction and Structural Analysis of α-Hairpinins, a Ubiquitous Family of Antimicrobial Peptides, Using the Cysmotif Searcher Pipeline

**DOI:** 10.3390/antibiotics13111019

**Published:** 2024-10-30

**Authors:** Anna A. Slavokhotova, Andrey A. Shelenkov, Eugene A. Rogozhin

**Affiliations:** 1Central Research Institute of Epidemiology, Novogireevskaya Str., 3a, 111123 Moscow, Russia; 2Shemyakin and Ovchinnikov Institute of Bioorganic Chemistry RAS, Miklukho-Maklaya Str., 16/10, 117437 Moscow, Russia; rea21@list.ru; 3All-Russian Institute for Plant Protection, Podbelskogo Str., 196608 Saint-Petersburg-Pushkin, Russia

**Keywords:** antimicrobial peptide, plant defense, α-hairpinin, plant transcriptome

## Abstract

Background: α-Hairpinins are a family of antimicrobial peptides, promising antimicrobial agents, which includes only 12 currently revealed members with proven activity, although their real number is supposed to be much higher. α-Hairpinins are short peptides containing four cysteine residues arranged in a specific Cys-motif. These antimicrobial peptides (AMPs) have a characteristic helix−loop−helix structure with two disulfide bonds. Isolation of α-hairpinins by biochemical methods is cost- and labor-consuming, thus requiring reliable preliminary in silico prediction. Methods: In this study, we developed a special algorithm for the prediction of putative α-hairpinins on the basis of characteristic motifs with four (4C) and six (6C) cysteines deduced from translated plant transcriptome sequences. We integrated this algorithm into the Cysmotif searcher pipeline and then analyzed all transcriptomes available from the One Thousand Plant Transcriptomes project. Results: We predicted more than 2000 putative α-hairpinins belonging to various plant sources including algae, mosses, ferns, and true flowering plants. These data make α-hairpinins one of the ubiquitous antimicrobial peptides, being widespread among various plants. The largest numbers of α-hairpinins were revealed in the *Papaveraceae* family and in *Papaver somniferum* in particular. Conclusions: By analyzing the primary structure of α-hairpinins, we concluded that more predicted peptides with the 6C motif are likely to have potent antimicrobial activity in comparison to the ones possessing 4C motifs. In addition, we found 30 α-hairpinin precursors containing from two to eight Cys-rich modules. A striking similarity between some α-hairpinin modules belonging to diverse plants was revealed. These data allowed us to assume that the evolution of α-hairpinin precursors possibly involved changing the number of Cys-rich modules, leading to some missing middle and C-terminal modules, in particular.

## 1. Introduction

During the last decade, humanity faced the problem of antibiotic resistance of pathogenic and opportunistic bacteria and the reduction in the effect of using antibiotics to overcome the infections caused by these bacterial agents. According to the study published in *The Lancet*, in the year 2019, nearly 1.3 million people died from diseases caused by antimicrobial-resistant pathogens, while the total number of fatal cases associated partly with antimicrobial resistance (AMR) was estimated to be about 5 million [1]. Consequently, searching for novel antimicrobial agents with various types of action became one of the major challenges in human healthcare.

Antimicrobial peptides (AMPs) have recently been a point of a special interest as naturally occurring molecules with a broad spectrum of antifungal, antibacterial, and antiviral activities [2,3]. These small proteins have been found in almost all living organisms from bacteria to eukaryotes [4] and have been considered as the innate immunity components, serving as the first line of defense [5]. Being widely diverse, AMPs share some common features, including small size, positive surface charge, and amphiphilic structure [6]. Numerous studies showed that many AMPs interact with membranes of pathogens and induce membrane permeabilization, thus killing the microorganisms [7]. This mechanism of action suggests small risks of AMR developing, making AMPs a prospective source of natural antibiotics [8].

Plant AMPs were first discovered in the early 1970s [9], and currently, their number exceeds 800 [10]. The majority of plant AMPs have a compact spatial structure due to the presence of several cysteine residues forming the so-called cysteine motif (Cys-motif) [11]. The structure of many AMPs was previously shown to be stabilized by disulfide bonds, and cysteines play an essential role in such bond formation [12,13]. According to Cys-motif and some other characteristics, AMPs are divided into several families, including a small family of α-hairpinins [14]. The latter are small peptides with a very specific Cys-motif that can be denoted as C^1^X_3_C^2^X_n_C^3^X_3_C^4^, where C^i^ shows cysteine residues (i = 1…4), X can independently be any amino acid residue except cysteine, and subscripts 3 and n indicate the number of such non-cysteine residues. This cysteine arrangement facilitates a common helix−loop−helix spatial structure comprising two α-helices oriented antiparallel and joined by a loop (Figure 1) [14]. Motif sequences can be highly heterogeneous, even within the same plant species, with cysteines sometimes being the only conservative residues, a fact that significantly complicates the possibility of detecting a new α-hairpinin using a homology search only.

It should be noted that until recently, α-hairpinins were considered to contain only four cysteines. However, according to the primary Cys-motif of the precursors, the fifth or sixth cysteines located one to three amino acids apart from the main ones can also be presented. Moreover, novel antibacterial α-hairpinin with six cysteines has been recently discovered in a ghost pepper [15]. In addition, α-hairpinin with six cysteines and the characteristic helix−loop−helix structure was isolated from *Nigella sativa* (Figure 1, PDB ID: 2NB2, https://www.rcsb.org/structure/2nb2, accessed on 29 September 2024). Thus, we can conclude that α-hairpinins are peptides with the specific 4–6 cysteine residue motif sharing common helix−loop−helix structure (Figure 1).

Representatives of this family exhibit a broad spectrum of activity [14]. For example, several α-hairpinins can bind a trypsin used by insects for protein digestion and inhibit its proteinase activity [16,17,18]. Some α-hairpinins display antiviral activity; in particular, they can inhibit protein synthesis of invaders and possess strong ribosome-inactivating activity [19,20]. Finally, the majority of α-hairpinins show strong antifungal activity against a wide range of fungi [14]. Some of these peptides were also active against bacterial pathogens [21]. The mechanism of α-hairpinin action was studied on the antifungal peptide EcAMP1, which exhibited activity via initial binding with components of a fungal cell wall, glycoproteins, proteins–amyloids, and glycans, in particular. After binding, EcAmp1 was distributed evenly over the cell surface and interacted with the plasma membrane, which provoked the peptide internalization into the cell surface and presumably induced apoptosis [22].

Biochemical isolation of plant AMPs is always time- and labor-consuming since acidic extraction followed by multistage stepwise chromatography is usually used [23]. On the other hand, a wide range of transcriptomic and genomic data of diverse plant species are currently available in public databases. Therefore, it is highly preferable to use these data for AMP prediction. There are many platforms for a novel AMP search (e.g., ACEP [24], amPEPpy [25], AmPEP [26], sAMPpred-GAT [27]), most of which are based on machine-learning methods. However, they usually lack specificity in their predictions and do not allow the revealing of the peptides belonging to a particular class. A recent study highlighted an existing bias in AMP predictors when dealing with disordered regions within AMP sequences and considered this as a specificity-limiting factor [28]. To the best of our knowledge, no existing algorithm is able to correctly predict α-hairpinins in wide-scale analysis.

Recently, we developed the Cysmotif searcher computational pipeline for AMP predictions in silico [29]. Using this software, several plant transcriptomes were analyzed to reveal their specific AMP profiles [30,31]. In the current study, we developed a novel algorithm for the prediction of α-hairpinin peptides. This algorithm was later integrated into Cysmotif searcher software and is now available for users in Github (https://github.com/fallandar/cysmotifsearcher, accessed 20 October 2024). We analyzed more than 1200 transcriptomes available in the One Thousand Plant Transcriptomes project (1KP) [32] that resulted in the prediction of more than 2000 putative α-hairpinins. Besides classification of putative α-hairpinins and their distribution among plant families, the diversity of α-hairpinin precursors was also analyzed. Below, we will refer to all predicted α-hairpinin-like peptides as ‘α-hairpinins’ for the sake of brevity, although not all of them might exhibit antimicrobial activity in vitro and in vivo.

We believe that the data obtained will facilitate the isolation and activity confirmation for this AMP family, which can ultimately lead to the discovery of potent antimicrobial agents against various plant and, possibly, human pathogens.

## 2. Results

### 2.1. Prediction of α-Hairpinin Peptides Using the 4C Cys-Motif

A general α-hairpinin motif, C^1^X_3_C^2^X_4-20_C^3^X_3_C^4^, was deduced from the primary structure of α-hairpinins with proven antimicrobial activity and named 4C by us. It includes three non-cysteine amino acid residues between the first and the second cysteines, as well as between the third and fourth ones. The number of residues between the second and the third cysteines in the motif can vary from 4 to 20. For example, both sequences **C^1^**AVR**C^2^**KLTM**C^3^**VRD**C^4^** (Seq. 1) and **C^1^**WMP**C^2^**SLQPD**C^3^**LTW**C^4^** (Seq. 2) possess the motif generalized above, although their similarity is low, and they have different numbers of residues (four and five, respectively) between the second and the third cysteines. At the same time, a sequence **C^1^**AVRC**^2^**KLT**C^3^**VRD**C^4^** (Seq. 3) does not contain the motif, although it is highly similar to Seq. 1, since it includes three residues between the second and the third cysteines, and this number is not within the range of 4–20.

The motif provided above was used for an in silico AMP search in transcriptomes available from the 1KP [32]. As a result, 1269 predicted α-hairpinins were found (Figure 2A, Appendix A). Among them, 869 contained four cysteine residues arranged in the 4C motif that varied by different numbers of amino acids between the second and the third cysteines shown as ‘X_n_’. The top five n values were 6, 4, 13, 5, and 7 occurred in 106, 92, 81, 78, and 68 predicted α-hairpinins, respectively (Figure 2C).

Considerably less predicted AMPs (pAMPs), namely, 289, had five cysteines in their primary structure. Among them, 84 detected peptides possessed a 4C+1 motif, which meant that one more cysteine located further than four amino acids ahead of the C1 was present in addition to the ‘classic’ 4C motif. The n values for this motif are shown in Figure 2A. A total of 205 α-hairpinins had a motif named 5C (C^5^X_1–3_C^1^X_3_C^2^X_4–20_C^3^X_3_C^4^). This motif was separated from the 4C+1 by us since the fifth cysteine located so close to the first one might be involved in the formation of an elongated α-helix. The most frequent were 5C putative peptides with eight amino acids between the second and the third cysteines, and their number was 93; other n values are indicated in Figure 2D and Appendix A.

Finally, 111 α-hairpinins predicted in silico had six cysteines, including 95 pAMPs with 4C+2 motifs. Putative 4C+2 peptides possessed two additional cysteines located further than three amino acids apart from C1 besides the basic 4C motif (Appendix A). Among these predicted α-hairpinins, the most common motif was C^6^X_2_C^5^X_30_C^1^X_3_C^2^X_7_C^3^X_3_C^4^, which occurred in 68 peptides. The other 16 predicted peptides were arranged to a group sharing the 5C + 1 motif that contained one additional cysteine to the main motif 5C (Appendix A).

The distribution of 1269 α-hairpinins among plant families and groups, as well as their expression in various plant organs, are reported in Figure 2B. These pAMPs were mostly found in plants belonging to the *Papaveraceae* family (73 predicted peptides); the second abundant family was, surprisingly, *Chlamydomonadaceae* (48 pAMPs), followed by *Poaceae* (46 pAMPs) and *Fabaceae* (40 pAMPs). *Cupressaceae*, *Onagraceae*, *Lamiaceae*, *Asteraceae*, *Pteridaceae*, and *Linaceae* had from 34 to 20 predicted α-hairpinins, while other families had less than 20 predicted peptides. The predicted α-hairpinins belonged mainly to dicots (607 pAMPs), algae (267 pAMPs), ferns (168 pAMPs), and monocots (118 pAMPs). Significantly smaller numbers of pAMPs were revealed in mosses (43), horn- and liverworts (43), lycophytes (22), and eusporangiate monilophytes (16), while less than 10 α-hairpinins were found in Euglenozoa, basalmost angiosperms, *Dinophyceae*, *Gnetales*, and *Cycadales* families. Predicted α-hairpinins were mainly expressed in leaves (548 pAMPs), algae cells (195 pAMPs), flowers (137 pAMPs), shoots (128 pAMPs), stems (50 pAMPs), and roots (21 pAMPs) (Appendix A).

The sequence logo for the most widespread 4C motif, C^1^X_3_C^2^X_6_C^3^X_3_C^4^, is shown in Figure 3. There were no dominating residues in non-cysteine positions, except for slightly prevalent serine (S) in the eighth position.

Furthermore, we analyzed the sequences of the predicted α-hairpinins in order to check whether they can possess antimicrobial activity with antifungal and antibacterial activity prediction web-servers. The results of this analysis are presented in Appendix A. In total, 83% of the peptides found were predicted to possess antifungal or antibacterial activity, or both. However, these results are greatly affected by changing the possible proteolysis site, which is rather hard to reliably detect in silico but can alter the length of a resulting peptide and, in turn, its predicted activity. Thus, the real number of 4C peptides with antifungal activity can be even higher.

### 2.2. Prediction of α-Hairpinins Using the 6C Cys-Motif

In addition to the basic 4C α-hairpinin motif, a novel one with six cysteines was recently discovered. This motif, denoted 6C (C^1^X_3_C^2^X_3_C^3^X_4-20_C^4^X_3_C^5^X_3_C^6^), was used for in silico search within 1KP plant transcriptomes. Surprisingly, the 6C motif was widely distributed, with 1069 pAMPs assigned to this second class of putative α-hairpinins. In particular, 1057 of them had six cysteines, while the remaining 12 included one or two cysteines in addition. Among the 1057 peptides, the most frequent were those with 10, 9, 12, 11, 6, and 8 amino acids between the third and the fourth cysteines observed in 558, 134, 110, 64, 54, and 51 pAMPs, respectively (Figure 4A, Appendix A). Considerably less predicted peptides contained 13, 4, and 7 amino acids in these positions (29, 18, and 13 pAMPs, respectively), and less than 10 pAMPs were contained in each group of peptides with 5 or 14–18 amino acids between the third and the fourth cysteines. Concerning the predicted peptides with additional cysteine residues, eight of them had the seventh cysteine located 42 amino acids apart from the 6C motif, while two pAMPs had a more compact motif and contained eight amino acids between the seventh cysteine and the main motif (Appendix A). Moreover, two predicted α-hairpinins had eight cysteines in their Cys-motif, which were located close to each other (C^8^X_4_C^7^X_2_C^1^X_3_C^2^X_3_C^3^X_8_C^4^X_3_C^5^X_3_C^6^).

We also studied the distribution of pAMPs with 6C motifs among plant taxons together with their expression in different parts of the plant. Similarly to 4C α-hairpinins, the 6C pAMPs were abundant in the members of the *Papaveraceae* family that contained 162 predicted peptides (Figure 4B, Appendix A). A total of 58, 53, 30, 29, 26, and 22 pAMPs were detected in *Onagraceae*, *Solanaceae*, *Boraginaceae*, *Fabaceae*, *Apocynaceae*, and *Lamiaceae*, respectively, which was significantly less than in *Papaveraceae*. Interestingly, besides the listed families, as many as 143 families contained from 2 to 15 α-hairpinins with the 6C motif (Appendix A). The described motif was found predominantly in dicots and was much less common, e.g., in monocots (about 60 pAMPs) and mosses (22 pAMPs) (Appendix A). The predicted 6C α-hairpinins were detected in different plant organs; the most abundant were leaves (more than 500 pAMPs in total), flowers and fruits (almost 300 pAMPs), and shoots (118 pAMPs), while roots contained considerably less AMPs (Appendix A).

In total, 95% of the revealed 6C peptides were predicted to possess antimicrobial activity, mostly antifungal, with third-party prediction servers. The activity prediction results are shown in Appendix A.

A sequence logo for the most widespread 6C motif, C^1^X_3_C^2^X_3_C^3^X_10_C^4^X_3_C^5^X_3_C^6^, is shown in Figure 5. In this case, the prevalence of particular non-cysteine residues was more prominent than for the 4C motif. For example, charged residues aspartic acid (D), arginine (R), and lysine (K) dominated in the 3rd, 13th, 15th, 19th, 22nd, 23^rd^, and 27th positions, while polar threonine (T) and hydrophobic valine (V) held 7th and 10th positions, respectively.

### 2.3. Prediction of Precursor Proteins with Several α-Hairpinin Domains

A special algorithm predicting α-hairpinin precursor proteins with more than one α-hairpinin module was developed. As a result, 30 precursors were identified with a number of α-hairpinin modules varying from 2 to 8. A total of 16 precursors had two AMP modules, including five relatively short putative precursors in which α-hairpinin sequences were separated by spacers and had short C-terminal regions at their ends (group 2M α-hairpinins; Appendix A). Another 10 predicted precursors possessed a structure typical for vicilin seed storage proteins: variable N-terminal part with two α-hairpinin modules followed by N- and C-terminal cupin 7S vicilin-like domains (group 2M vicilin-like α-hairpinins; Appendix A). One more predicted precursor had two α-hairpinin domains on the opposite termini separated by a large middle part exhibiting similarity only to hypothetical proteins with unknown functions (group 2M termini α-hairpinin; Appendix A).

Besides two-modular α-hairpinin precursors, eight precursors with three domains, four with four domains, one with seven domains, and one with eight domains were detected in this work. Among them, four three-modular precursors contained also two cupin 7S vicilin-like domains (Appendix A), while another four three-modular precursors, as well as precursors with four, seven, and eight α-hairpinin modules, did not include these vicilin-like domains and were not terminated by the short C-region (Appendix A).

A total of 19 out of 30 predicted prepropeptides exhibited similarity to vicilin proteins; however, 6 of them did not contain cupin 7S vicilin-like domain (Appendix A). These six putative peptides, as well as some other α-hairpinin precursors available in GenBank, were incorrectly annotated as vicilin seed storage proteins, when in fact they had a typical modular structure. Moreover, two detected precursors exhibited similarity to antimicrobial peptides from various Poacea plants, while another two precursors had high homology with MBP-1 α-hairpinin [21] previously isolated from corn (*Zea mays*; Appendix A). Seven predicted proteins had considerable similarity only with uncharacterized proteins, including a precursor from *Anemone hupenhensis* with two closely located Cys-rich domains having 6C and 5C motifs simultaneously.

## 3. Discussion

### 3.1. In Silico Prediction of More Than 2000 α-Hairpinins

Until recently, it was thought that the family of α-hairpinins was a small group of AMPs including only 12 representatives with proven antimicrobial activity [14,15,33]. All these peptides were isolated from different plants by basic procedures including extraction and stepwise purifications. In this work, we applied a completely different approach and developed a special algorithm for in silico α-hairpinin prediction. Two Cys-motifs, namely, 4C and 6C, were used for the detection of α-hairpinins in the transcriptomes available in the 1KP project [32]. Surprisingly, more than 2000 putative α-hairpinins were found. Our pipeline relies on a conserved pattern search followed by several filtration steps, and it does not involve 3D structure prediction or antimicrobial activity verification. Nevertheless, most of the α-hairpinins revealed were found to possess antimicrobial activity by third-party prediction software (83% for 4C and 95% for 6C putative peptides, respectively). Thus, Cysmotif searcher should be used carefully, taking into account the described limitations, as with any other in silico prediction software.

It is worth noting that our pipeline has been recently used to reveal a novel 6C α-hairpinin peptide from ghost pepper [15]. The authors used the motifs developed by us as a starting point for revealing novel CC-AMP1-like α-hairpinin and confirmed its activity against bacterial pathogens from the ESKAPE group in vitro. In another publication, the authors used Cysmotif searcher for α-hairpinin mining in the genome of lima bean and revealed the PlHrp1 peptide, which was shown to possess antifungal and antibacterial activity using in silico analysis [34]. These recent discoveries confirm the possibility of Cysmotif searcher application for α-hairpinin mining in plant transcriptomes and genomes.

### 3.2. α-Hairpinins Are Ubiquitous Components of the Plant Defense System

In this study, more than 1200 4C and 1000 6C α-hairpinins were predicted, which completely changed our notions regarding this family. First, α-hairpinins seem to be ubiquitous, since they were observed in more than 200 plant families and occurred in various plant taxons from primitive plants and algae to true flowering plants such as *Asterids* and *Rosids*. Interestingly, the family most abundant with α-hairpinins was *Papaveraceae,* which includes numerous latex-bearing plants rich in different bioactive compounds such as alkaloids, carotenoids, phenols, and terpenoids [35]. Secondly, our previous study detected about 4000 putative defensins and 3000 putative lipid-transfer proteins [36] within 1KP, being known as one of the major components of plant immunity [37,38], and their number was comparable to the one of α-hairpinins found in this work. These data together with various types of α-hairpinin antimicrobial activity may point to the unique role of α-hairpinins in plant defense. Thirdly, one particular plant (*Papaver somniferum*) contained more than 40 various α-hairpinins with different 4C and 6C motifs (Appendix A). It should be mentioned that *Papaver somniferum* is an underestimated source of plant AMPs; although it is abundant with α-hairpinins (shown in this study) and thionins [39], we failed to find any information regarding peptides with proven antimicrobial activity isolated from this plant. In order to facilitate the investigations in this field, our study revealed a collection of α-hairpinins with 4C and 6C motifs, so that particular peptides can be selected and then expressed in vitro for the further investigation of their antimicrobial activities, spatial structures, or mechanisms of action.

The chi-squared test allowed us to reveal that the distribution of α-hairpinins among groups (e.g., core eudicots, conifers), families, and organs of plants was not as expected according to the frequencies of the corresponding cohorts in the initial 1KP dataset with α-values lower than 10^−10^. The numbers for particular species were too low to perform the test. The most significant differences of expected and observed values were found for basal eudicots (85 motifs vs. 56 expected), core eudicots (76 vs. 114), and green algae (205 vs. 166). For the families and organs, the most significant differences were revealed for *Asteraceae* (21 vs. 41 expected), *Chlamydomonadaceae* (48 vs. 22), and *Papaveraceae* (73 vs. 33), as well as for leaves (sum for all categories including ‘leaves’ or ‘leaf’: 398 vs. 501 expected). However, the only reliable conclusion found in these data could be a confirmation of *Papaveraceae* as a source of α-hairpinins.

We believe that the number and variety of predicted α-hairpinins in plants also indicates a significant role of these peptides in plant immunity. Furthermore, our results are in good coincidence with other studies devoted to AMP repertoire prediction. For example, a transcriptome of healthy *Stellaria media* seedlings contained 18 predicted α-hairpinin transcripts, while 12 such sequences were found in the transcriptome of shoots infected with *Fusarium oxysporum* [31]. Other examples were transcriptomic studies of *Peltophorum dubium* and *Leymus arenarius* seedlings that contained 10 and 16 α-hairpinin sequences, respectively [30,40]. At the same time, α-hairpinins were not found in some transcriptomes, although the goal of a study was to reveal the maximum numbers of defense peptides [41]. It is also worth noting that α-hairpinins were found in various plant organs: putative peptides with 4C motifs mostly occurred in leaves, algae cells, and flowers, while the majority of 6C putative peptides were observed in flowers and fruits, leaves, and shoots.

### 3.3. The Diversity of Detected Cys-Motifs

Another focus of this study was an analysis of Cys-motif diversity presenting among the predicted α-hairpinins. All the putative peptides were divided into two large classes possessing either the 4C or 6C motif. The first finding was that putative 6C peptides were more uniform than 4C peptides. In particular, 1057 out of 1069 6C α-hairpinins shared the same 6C motif and contained only six cysteines. In comparison, 4C α-hairpinins could be divided into five groups including the largest one with putative peptides sharing the 4C motif, as well as considerably smaller groups of predicted α-hairpinins with five and six cysteines. We speculate that additional cysteine residues may be involved in the formation of coils in α-helices, shorten unstructured tails, and provide the peptide with enhanced stability. Similar peptides with one or two additional cysteines were detected earlier in some modules of precursors of Sm-Amp-X, Tk-Amp-X [14], and MiAMP2c [42], and the presence of mature peptides with elongated Cys-motifs was confirmed by mass spectrometry and biochemical isolation [42]. The next assumption is that the 6C α-hairpinin class most likely contains more active AMPs than the first one. The prediction of antimicrobial activity in silico proved this assumption, since the fraction of 6C peptides possessing antifungal and/or antibacterial activity was higher than the one for 4C peptides (95% versus 83%, respectively). This conclusion also coincides with ‘n’ values and their frequencies among putative peptides. Previously known α-hairpinins with proven antimicrobial activity had the distance between the second and the third cysteines equal to 11–13 amino acids [14], but we detected a number of 4C peptides with 4–7 residues in this segment. At the same time, almost 900 putative 6C peptides had n values equal to 9–13 amino acids. In comparison, the number of 4C peptides with the same distance in the loop was only 250. Therefore, it seems plausible that considerably more peptides might be functional among 6C predicted α-hairpinins; they could occur as mature peptides in a plant and exhibit antimicrobial activity.

### 3.4. α-Hairpinin Precursors Could Evolve by Changing the Number of Cys-Rich Modules

A total of 30 α-hairpinin precursors containing from two to eight Cys-rich modules were predicted in this work using a specially developed algorithm. It is known that α-hairpinin precursors are divided into two types depending on the presence of the vicilin-like region [14]. Indeed, both of the types were found in the current study. The first type of precursors, termed vicilin-like, consists of a signal peptide, several Cys-rich α-hairpinin modules, and a vicilin-like C-terminal region [14,42,43]. Due to proteolysis, Cys-rich α-hairpinin modules are released from prepropeptide, become mature, and acquire their antimicrobial activity [14,44]. The hydrophobic C-terminal part serves as a seed storage protein and contains two cupin-7S vicilin domains that are peculiarly widespread within legumes, playing a role in sucrose binding and oxidative stress response, and they are also widely known as major food allergens [45]. We detected 13 vicilin-like precursors with two cupin-7S vicilin domains and two or three α-hairpinin modules.

The second type of precursor, named modular, usually exhibits no similarity to vicilins and consists of a signal peptide followed by a cassette with up to 12 α-hairpinin modules, and it ends with a short C-terminal domain [14]. Again, mature α-hairpinins start to function as antimicrobial peptides upon proteolysis of the precursor. Remarkably, this modular type of precursor is also typical for various antimicrobial peptides from animals including spiders [46,47] and scorpions [48,49]. We detected 17 plant modular precursors containing from two to eight α-hairpinin motifs.

We analyzed putative α-hairpinin precursors using multiple alignments of the whole precursors or their separated Cys-rich modules. As a result, two main conclusions were drawn. First, different plants belonging to distinct families can contain highly similar α-hairpinin modules or even whole precursors, and possibly have one common AMP ancestor. Second, the diversity of α-hairpinin precursors found in this work or those deposited to GenBank is based on changing the number of α-hairpinin modules (Figure 6). Specifically, N-terminal Cys-rich modules are mainly similar, while some middle and C-terminal modules can be removed.

These conclusions are common for both vicilin-like and modular α-hairpinin precursors. In particular, two highly similar Cys-rich modules of vicilin-like precursors isolated from plants belonging to primitive fern (*Danaea nodosa*), true flowering *Caryophyllaceae* (*Silene latifolia*), and *Acteraceae* (*Matricaria matricarioides*) families were observed (Appendix A). Then, N-terminal α-hairpinin modules of a succulent (*Delosperma echinatum*), a parasitic plant (*Orobanche fasciculate*), and plants from *Caryophyllaceae* (*Saponaria officinalis*) and *Lamiaceae* (*Lavandula angustifolia*) families were very similar as well (Appendix A). Moreover, these precursors exhibited rather high similarity throughout the whole protein length, which supports the idea of one possible common precursor for some vicilin-like prepropeptides. In modular α-hairpinin precursors, three out of five Cys-rich motifs of *Setaria italica* (GenBank ID XP_022685044; Appendix A) were absolutely identical and the other two differed by single amino acid substitutions. Highly similar to them were modules from Australian cereals *Thyridolepis multiculmis* and *Neurachne minor* predicted in this study (Appendix A). When their whole precursors were aligned, the high similarity was observed between the first two domains together with 13 C-terminal amino acids, while C-terminal Cys-rich modules were missed in precursors from *T. multiculmis* and *N. minor* (Figure 6 and Appendix A). This finding confirms the idea of developing α-hairpinin variety by removing some C-terminal Cys-rich modules. One more discovery supporting this assumption was the revelation of precursors from *Pycnanthemum tenuifolium* and *Microstegium vimineum* with three and seven Cys-rich motifs, respectively, which were similar to MBP-1 containing eight modular prepropeptides. Precursors from *P. tenuifolium* had only four mismatches among 166 amino acids (Appendix A), while precursors from *M. vimineum* contained considerably more substitutions. However, it exhibited high similarity along the whole precursor length and contained 11 identical C-terminal amino acids, while lacking the last eighth domain (Figure 6).

We also analyzed some cereal AMP precursors available in GenBank or described in the papers, and again found that within the cassette, some modules close to C-terminus can be removed, while in N-terminal regions, the last Cys-rich motif and C-terminal region remained intact. For example, *Sorghum bicolor* contains three different α-hairpinin precursors with 7–9 Cys-rich modules. The sixth and the seventh modules could be omitted when forming middle and short forms of precursors, while the last two Cys-rich motifs are presented in all isoforms (Figure 6 and Appendix A). Tk-Amp-X precursors isolated from *Triticum kiharae* [14] had a similar organization; in particular, the long form contained seven α-hairpinin modules, while short and middle precursors had reduced cassettes, in which the fifth and/or the sixth C-terminal modules were excluded (Figure 6). Moreover, precursor prepropeptide from *Echinochloa crus galli* [14], which was highly similar to eight-modular MBP-1, included a truncated cassette without the seventh domain (Figure 6). The same was also found in some vicilin-like α-hairpinin precursors. In particular, the precursors from *D. echinatum*, *O. fasciculate*, *S. officinalis*, and *L. angustifolia* mentioned above contained a quite similar N-terminal signal peptide, two first α-hairpinin modules, and two cupin-like domains, while the third Cys-rich motif was present only in a precursor from *L. angustifolia* (Appendix A).

Remarkably, during BLAST annotation, we many times observed the proteins similar to our predicted precursors, which possessed additional middle or next-to-last modules; however, we did not find any α-hairpinin precursors with a modified order of modules. We may speculate that elongation of α-hairpinin precursors by the addition of some Cys-rich modules led to increasing antimicrobial activity since more mature AMPs will be released from elongated precursor upon proteolysis, which, in turn, will lead to increasing the number of active AMPs and amplification of the antimicrobial activity as a whole, helping the plant to resist pathogen invasion. Arguably, proteolysis of different modules can be dependent on a stage of plant development reaching maximum in immature seeds and during seedling development, which are the most vulnerable stages.

### 3.5. The Limitations of the Current Study

The current study represents an in silico investigation of α-hairpinin features that can be used for the prediction of putative peptides from this family in plant transcriptomic sequences. Although the motifs used for such a prediction underwent careful curation using rather stringent criteria, it should be mentioned that unambiguous and specific in silico identification of α-hairpinins is a challenging task. Thus, the putative α-hairpinins predicted by Cysmotif searcher should be subjected to an additional selection process according to specific investigation goals before they will be ready for experimental verification. The investigators with a significant background in the field of plant AMP isolation and verification can readily propose additional filtration steps for the predicted peptide set to achieve better specificity or applicability in a particular case.

Another limitation is that we used the plant transcriptome dataset from the 1KP project for our prediction, and more recent and/or more specific datasets can provide slightly better results. In addition, our goal was to capture the general patterns for the α-hairpinin family and not to predict as many potentially active peptides as possible. For this reason, we excluded rarely occurring motifs from the downstream analysis, but they can potentially represent more specific cases and thus be of interest in some future investigations. The list of these single motifs can be found in Appendix A.

To conclude, α-hairpinins predicted by Cysmotif searcher provide a good starting point for future in vitro and in vivo activity investigations, but the limitations described above should always be taken into account when performing such analyses.

## 4. Materials and Methods

### 4.1. Development of the Criteria for α-Hairpinin Prediction

Most part of AMPs represent cysteine-rich peptides possessing a special type of signature sequences called cysteine motifs. The structure of such motifs is rather flexible, and sequence similarity between the motifs of a particular AMP family could be rather low, so that sometimes only cysteine residues have conserved positions in them. This makes their searching and classification a complex task when using homology-based methods only. Previously, we developed the computational pipeline ‘Cysmotif searcher’ [29] for revealing such motifs and classifying the peptides containing them into AMP families. The motifs used in the pipeline were deduced from literature data and AMP databases and manually curated by us to achieve the required sensitivity and specificity. The AMP families that could be revealed included defensins, thionins, cyclotides, snakins, hevein-like peptides, and lipid-transfer proteins. We revealed more than 10,000 potential AMPs [36] in 1267 plant transcriptomes from the 1KP project [32].

However, an important AMP class of α-hairpinins [14] was not covered by the motifs developed at that time due to insufficient data available. In order to reveal α-hairpinin sequences, we deduced several motifs using the verified α-hairpinin data currently available in public databases and the literature. The list of motifs is shown in Appendix A.

The difficulty of predicting α-hairpinins lies in the fact that their Cys-motifs are widespread and could be a part of Cys-motifs belonging to other AMPs such as defensins, lipid-transfer proteins, and thionins. For this reason, all the predicted peptides were additionally manually curated and annotated by BLAST search, and only the sequences that have passed all filtration steps were characterized as putative α-hairpinins. The curation criteria for inclusion and exclusion of the peptides from α-hairpinin class are shown in Table 1.

Inclusion criteria were the presence of either the 4C or 6C motifs. The possibility of a presence of one or two additional cysteines before or after the motif was based on the reports that some precursors of α-hairpinins contained additional cysteines, which can be located either quite close to the source 4C/6C motif or at a significant distance from it.

Exclusion criteria were the absence of the 4C/6C motifs. Furthermore, the presence of additional cysteine residues beyond the α-hairpinin motif was also assessed. If double cysteines before or after the motif were present, this sequence was excluded, since this feature was a characteristic of other plant AMPs (thionins, hevein-like peptides, lipid-transfer proteins etc.). Moreover, if there were three or more cysteines outside the core motifs, they had to be arranged in an additional α-hairpinin motif or its part, otherwise this sequence was excluded. The reason for such exclusion is that there are some plant AMP families that have 4C/6C core motifs as a part of more broad Cys-motifs, but other cysteines in such motifs have a different arrangement. Another exclusion criterion was the presence of X_1_CX_3_C immediately after the motif, since this is the characteristic C-tail sequence of defensins. Finally, if BLAST annotation characterized a sequence as representing non-hairpinin protein (e.g., thionin) except vicilins and unknown proteins, it was also excluded from further study.

We used the same set of 1267 plant transcriptomes as in our previous investigations [36] in order to make the results of two studies comparable with each other and exclude non-hairpinin motifs as described above.

### 4.2. Description of the Prediction Pipeline

Searching for motifs in translated transcriptomic sequences from the 1KP project was performed by Cysmotif searcher. The version used to obtain the results in this manuscript was 3.3.3 (source code is available at https://github.com/fallandar/cysmotifsearcher under GNU GENERAL PUBLIC LICENSE Version 3, accessed on 29 September 2024). Novel motifs are included in the file ‘motifs_hairpinin.txt’ available from the repository given above.

The flowchart of the pipeline is shown in Figure 7.

In general, the pipeline includes three stages, namely, prediction of the cysteine motifs, their filtration, and classification of the results. Cysmotif searcher accepts nucleotide or translated transcriptomic sequences and motif list as input, performs 6-reading frame translation when necessary, and then reveals open reading frames (ORFs) starting from methionine residues in resulting amino acid sequences. The presence of methionine provides additional evidence that these sequences are unlikely to be the artifacts. The transcripts are not subjected to homology- or similarity-based filtration to avoid the algorithm overfitting. In the next step, checking for the presence of the motifs from the provided list in the revealed ORFs is performed. If two motifs overlap each other, then the one containing greater number of cysteine residues is selected. In addition, no cysteines are allowed to appear after the motif revealed in the same ORF, except for the case of motif extension or searching for modular motifs, as described below. These steps are intended to select the most reliable motifs from the transcriptomes. Next, the amino acid sequences containing motifs are filtered based on the presence of a signal peptide in them (using SignalP 5.0 [50]), and the sequences not containing a signal peptide are excluded from further processing. Finally, the sequences are filtered based on the length of a mature peptide (<=600 aa), again to exclude possible artifacts. The peptides that have passed all the filters are classified based on the motifs revealed in them. In the case of α-hairpinin searching, the classification includes 4C and 6C classes.

An additional set of dedicated software (scripts) developed by us was used to collect the descriptive statistics on plant families possessing each type of motif. The script cysmotif_stat.sh can be found at Github ((https://github.com/fallandar/cysmotifsearcher, accessed on 29 September 2024). This dedicated script collects the motif distribution among plant groups, families, species, and parts. It takes the output file of the main pipeline and annotation file from 1KP project as input and distributes the sequences revealed into cohorts based on the annotation provided. The first level includes high-order groups (e.g., core eudicots), the second includes families, the third level is for species, and the fourth one is for tissues. The output is provided as text files in tabular form.

Upon obtaining a filtered and curated set of potential α-hairpinin sequences, we excluded the motifs revealed in only one sequence in order to increase the reliability of the analysis. Our main goal was to provide future activity determination experiments with more reliable and generalized data, not to capture the complete possible diversity of potential α-hairpinins. Single sequences can represent rare peptides, but they can also arise from sequencing errors or artifacts introduced by transcriptome assembly software. However, such sequences can be of interest for some researchers, so we added them to a separate Appendix A, but we did not include them in the downstream analysis since they were not supposed to reflect the general properties of the plant families or groups to which they belonged.

In order to check whether the peptides revealed could in fact possess antimicrobial activity against some fungi or bacteria, we submitted all the sequences obtained to prediction servers. Antibacterial activity was assessed by AntiBP3 [51] (https://webs.iiitd.edu.in/raghava/antibp3, accessed on 24 September 2024, threshold value = 0.5), and antifungal activity—by Antifp [52] (http://webs.iiitd.edu.in/raghava/antifp, accessed on 24 September 2024, threshold value = −0.3) and AfpTransferPred [53] (https://selectFigureht.org/afptransferpred/, accessed on 24 September 2024, threshold value = 0.5). In the latter case, a peptide was considered antifungal if both servers predicted this.

It is essential to verify the results of all in silico prediction procedures for transcriptomic sequences using some random shuffling algorithm to exclude possible artifacts [54]. For the purpose of additional verification of the prediction consistency and exclusion of possible false positives, the motif searching was additionally conducted in the amino acid sequences built with a random number generator. We took ORFs, in which the motifs were found, performed their random shuffling using the Fisher–Yates shuffle procedure, and subjected the obtained amino acid sequences to motif searching. Only two motifs corresponding to 4C peptides and no motifs corresponding to 6C peptides were revealed, which makes less than 0.2% of the corresponding numbers for the real transcriptomic sequences. This procedure confirms that the probability of revealing the exact motif structure by chance is very low.

### 4.3. Prediction of Modular α-Hairpinin Precursors

Another important addition to the pipeline was the procedure of revealing the sequences possessing ‘modular’ motifs, which included several repeats (up to 6) of a particular motif in the same ORF. These repeats could have similar, but not identical, sequences. This improvement was achieved by a semi-automated algorithm, the implementation of which is a script cysmotif_modular.sh available in Github (see link above). The algorithm includes running cysmotif_searcher.pl several times with the same set of motifs to search, but the output of the previous run becomes an input to the next run. The motifs revealed are marked in lower case in the corresponding amino acid sequence of a peptide, and cysmotif_searcher.pl is instructed with the –u option to skip conversion of input sequences to upper case. Such a trick, together with a processing logic set to report only the first one from two non-overlapping motifs within a single sequence, allows the program to ignore the previously revealed motifs and search for additional ones within the same sequence in order to reveal multiple domains in a peptide. The exemplary sequential pipeline calls are as follows:

cysmotif_searcher.pl -i AAAA_translated.fasta.bz2 -m motifs_hairpinin.txt -t -b -n 55 -l 600 –c;

cysmotif_searcher.pl -i AAAA_translated_motifs_orfonly_withM.fasta -m motifs_hairpinin.txt -t -b -n 55 -l 600 -c –u.

The description of all options can be found in Github (see above).

An example of a modular structure is provided in Appendix A. Previously, a sequence including additional cysteines beyond the motif was classified as belonging to the artificial ‘cysteine-rich peptide’ class and excluded from further analysis due to uncertainty regarding its tertiary structure [36].

### 4.4. Statistics Collection and Result Presentation

Sequence logos for 4C and 6C motifs were generated using the Weblogo resource (https://weblogo.berkeley.edu/logo.cgi, accessed on 20 October 2024).

Then, we collected the descriptive statistics on the motif distribution among plant families and parts of the plants using the cysmotif_stat.sh script described above. The full list of the peptide sequences revealed is available in Appendix A (multidomain), Appendix A (4C and 6C motifs), and Appendix A (4C and 6C sequences with motifs revealed only once, which were excluded from the downstream analysis).

We also performed statistical analysis using the chi-square test to check whether the distribution of the revealed α-hairpinins among various transcriptome groups (plant families, organs, etc.) simply reflects the frequency of such groups in the total number of transcriptomes or not. For example, our null hypothesis was that the distribution of α-hairpinins is independent of the plant family. We used the *p*-value of 0.01 as a threshold value in the test. To ensure the significance of the results and criterion applicability, we excluded the groups containing less than six sequences from the analysis.

## 5. Conclusions

In this study, we developed a novel algorithm for reliably detecting α-hairpinins in plant transcriptomic sequences and expanded it to reveal peptides possessing a complex modular structure, which cannot be predicted by a simple homology search. Several important findings were revealed in this study based on the application of this novel algorithm to the analysis of transcriptomic sequences from the 1KP project:i.being widely widespread among various plant families, α-hairpinins are especially prevalent in *Papaveraceae*, in particular, *Papaver somniferum*, which contains 43 various α-hairpinins with different Cys-motifs belonging to two large classes 4C and 6C;ii.more than 2000 putative α-hairpinins were predicted, among which the peptides with the 6C motif are more likely to possess strong antimicrobial activity and are more important to isolate by biochemical methods from the corresponding plants, at least according to in silico activity prediction;iii.the diversity of α-hairpinin precursors was possibly developed by changing the number of α-hairpinin modules; specifically, some middle and C-terminal Cys-rich modules can be removed without losing the peptide function;iv.the unambiguous identification of α-hairpinins represents a very difficult and challenging task, and thus a careful curation of the sequences obtained using the provided motifs is needed since their prediction was based solely on the transcriptomic sequences; researchers with a significant background in the field can propose their own curation and filtration steps, which will supplement the steps included in the Cysmotif searcher pipeline.

The data and software provided will facilitate the transcriptome-based mining and possible isolation of peptides with high antimicrobial activity from different plants, which can ultimately contribute to developing better antimicrobial drugs.

## Figures and Tables

**Figure 1 antibiotics-13-01019-f001:**
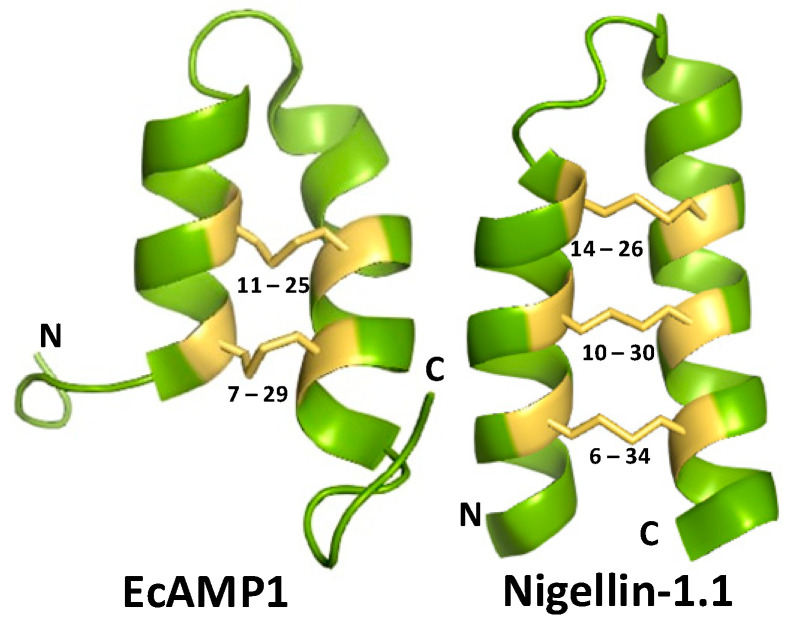
Examples of α-hairpinin spatial structures. On the left, Ec-Amp1 (PDB code 2L2R) with four cysteines is displayed, and on the right, Nigellin-1.1 (PDB code 2NB2) with six cysteines is shown. Disulfide bridges are shown as yellow sticks, cysteine residues are highlighted in yellow, and their positions are specified.

**Figure 2 antibiotics-13-01019-f002:**
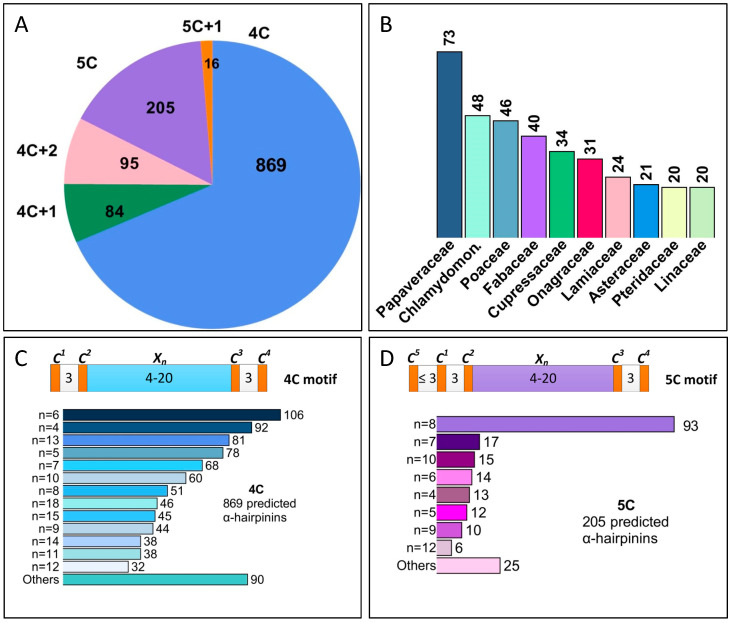
Diversity of the first class of putative α-hairpinins predicted using the 4C motif. (**A**) Classification of α-hairpinins depending on the Cys-motif. Outside of a pie chart, the designations of the motifs are shown (**4C**, C^1^X_3_C^2^X_4-20_C^3^X_3_C^4^; **4C + 1,** C^5^X_≥4_C^1^X_3_C^2^X_4-20_C^3^X_3_C^4^; **4C + 2,** C^6^XC^5^X_≥4_C^1^X_3_C^2^X_4-20_C^3^X_3_C^4^; **5C,** C^5^X_≤3_C^1^X_3_C^2^X_4-20_C^3^X_3_C^4^; **5C + 1,** C^6^X C^5^X_≤3_C^1^X_3_C^2^X_4-20_C^3^X_3_C^4^); inside the pie chart, the numbers display the amount of α-hairpinins with the corresponding Cys-motif. (**B**) The top 10 plant families including putative α-hairpinins with the 4C motif and its derivatives. (**C**,**D**) Schemes and bar charts of two prevailing Cys-motifs. Cysteine residues are displayed as orange boxes and signified as C^1-5^, where the superscripts designate a serial number in the motif. Numbers between cysteines independently denote any amino acids except cysteine; **Xn**, any amino acids except cysteine between the second and the third cysteines, independently. Bar charts display the number of predicted α-hairpinins with different n values.

**Figure 3 antibiotics-13-01019-f003:**
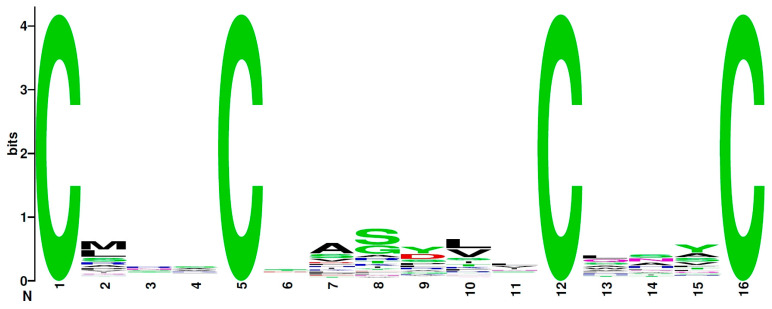
A sequence logo of the most prevalent 4C motif, C^1^X_3_C^2^X_6_C^3^X_3_C^4^.

**Figure 4 antibiotics-13-01019-f004:**
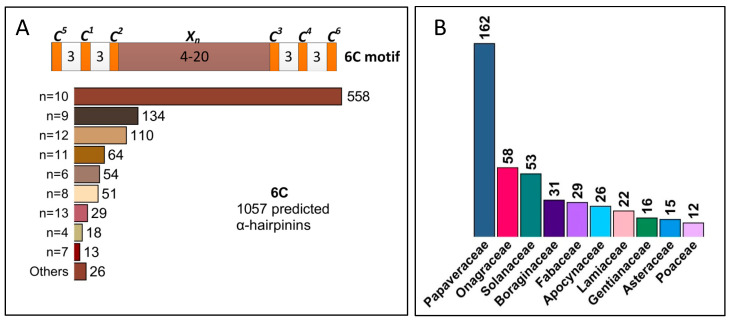
Variety of α-hairpinin-like peptides with the 6C motif. (**A**) A bar chart displaying a distribution of different n values (number of non-cysteine amino acids between C2 and C3) found in the predicted 6C motif-possessing peptides. (**B**) Top 10 plant families abundant with 6C putative α-hairpinins.

**Figure 5 antibiotics-13-01019-f005:**
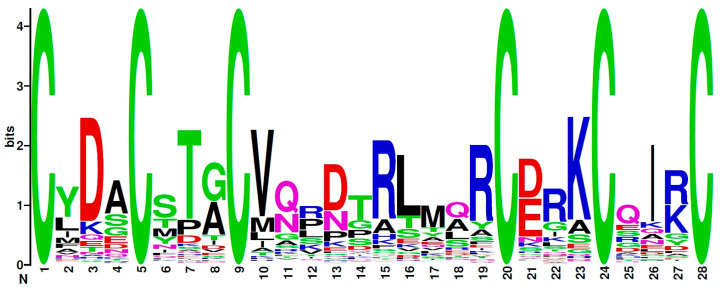
A sequence logo of the most prevalent 6C motif, C^1^X_3_C^2^X_3_C^3^X_10_C^4^X_3_C^5^X_3_C^6^.

**Figure 6 antibiotics-13-01019-f006:**
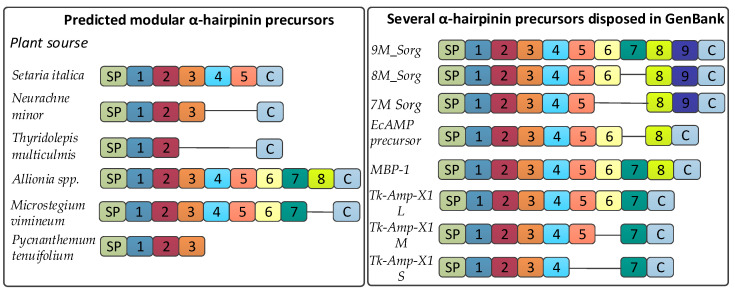
Modular α-hairpinin precursors differ by the amount of C-terminal Cys-rich motifs. SP, signal peptide; 1–8, α-hairpinin modules; C, C-terminal region. Precursor from *S. italica* has GenBank ID XP_022685044. Precursors from *N. minor*, *T. multiculmis*, *Allionia* spp., *M. vimineum*, and *P. tenuifolium* have the following designations in Appendix A: 3M_BXAY, 2M_WCOR, 8M_EGOS, 7M_YPIC, and 3M_DYFF. 9M_Sorg, 8M_Sorg, and 7M_Sorg are precursors with GenBank IDs XP_002449751.1, XP_021316824.1, and XP_021316825.1, respectively. MBP-1 is a precursor of antimicrobial peptide MBP-1 from *Z. mays* (NP_001142639). EcAMP precursor was deduced from partial sequences with GenBank IDs KF478770 and KF478781. Tk-Amp-X1 is proteolysed from L, M, and S precursors with Uniprot accession numbers HF562352, HF562351, and HF562347, respectively.

**Figure 7 antibiotics-13-01019-f007:**
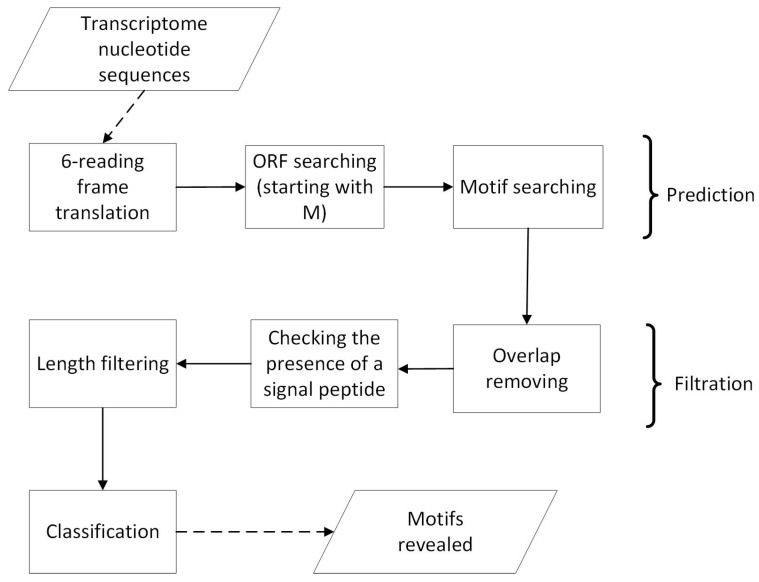
A flowchart of the Cysmotif searcher prediction pipeline operation.

**Table 1 antibiotics-13-01019-t001:** Inclusion and exclusion criteria for the manual curation of potential α-hairpinin peptides.

Inclusion Criteria	Exclusion Criteria
1. The presence of the 4C motif (CX_3_CX_4__–20_CX_3_C)	1. The absence of the 4C/6C core motif
2. The presence of the 6C motif (CX_3_CX_3_CX_4–20_CX_3_CX_3_C)	2. The presence of ‘CC’ before or after the 4C/6C core motif
3. The presence of one or two additional cysteines located before or after the 4C/6C core motif	3. The presence of three or more cysteines, which are not arranged in the 4C/6C core motif, before or after the core motif itself
	4. The presence of X_1_CX_3_C after the motif
	5. BLAST annotation of a peptide as belonging to some other protein family, except vicilins and unknown proteins

## Data Availability

The supporting data are available in Appendix A. Source code of the software used and user instructions are available on Github (https://github.com/fallandar/cysmotifsearcher, accessed 24 October 2024).

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
