# Peer review of "Computational Prediction and Structural Analysis of α-Hairpinins, a Ubiquitous Family of Antimicrobial Peptides, Using the Cysmotif Searcher Pipeline"

_antibiotics, 2024, doi:10.3390/antibiotics13111019_

Round 1
Reviewer 1 Report
Comments and Suggestions for Authors
The paper addresses one of the most challenging families of antimicrobial peptides, due to their complexity and structural variability. The authors have developed an interesting tool, but I would like them to clarify the following aspects:
In page 10 (lines 405-406) the authors affirm: “The AMP families that could be revealed included defensins, thionins, cyclotides, snakins, hevein-like peptides and lipid-transfer proteins.”
My question is: if α-hairpinins share cysteine ​​motifs with other AMPs, this suggests that classification as α-hairpinins may be based on the presence of these structural motifs, and not necessarily on a unique function or a distinct category of peptide. It would be important for the authors to clarify how they clearly differentiate α-hairpinins from other peptides that share these motifs.
In page 10 (lines 415-417), where the authors say: “For this reason, all the predicted peptides were additionally manually curated and annotated by BLAST search, and only the sequences that have passed all filtration steps were characterized as putative α-hairpinins.”
My concern is that the authors does not clearly detail the filtering steps after the BLAST search. To ensure the reproducibility and validity of their results, it is essential that the authors describe in detail all the criteria and methods used in filtering the sequences, including how the inclusion or exclusion of certain sequences was decided.
In page 10 (lines 418-420), where the authors affirm: “We used the same set of 1,267 plant transcriptomes as in our previous investigations [29] in order to make the results of two studies comparable with each other and exclude non-hairpinin motifs as described above.”
Again, if α-hairpin motifs appear in other proteins, such as defensins, the paper needs to clarify whether these peptides were considered as α-hairpins or excluded, especially considering that defensins (for instance) may not be detected y BLAST. The methodology should include a clear explanation of how they dealt with these overlaps to avoid ambiguity.
In page 10 (lines 431-432), Where the authors state: “Additional set of dedicated software (scripts) developed by us was used to collect the statistics on plant families possessing each type of motif.”
Where do the authors show these “additional scripts” or explain their specific functionalities? For a complete evaluation of the method, it would be necessary for the authors to detail what these scripts do, what kind of statistics are collected (such as motif frequency, distribution among species, etc.), as well as how this information was used in the analysis.
In page 11 (lines 439-442): “Previously a sequence including additional cysteines beyond the motif was classified as belonging to artificial ‘cysteine-rich peptide’ class and excluded from further analysis due to uncertainty regarding its tertiary structure [29].”
I think that including a conserved domain evaluation could enhance the robustness of the findings. Given the brevity of the motifs, such an analysis would help mitigate potential mistranslation biases and ensure the validity of the conclusions. Could the authors elaborate on how they addressed this potential bias in the study?
Page 11 (lines 445-446): “Then we collected the statistics on the motif distribution among plant families and parts of the plants. Full list of the peptide sequences revealed is available in Tables S1 (multidomain) and S2 (4C and 6C motifs).”
To facilitate a clearer understanding of the results, it would be beneficial to detail the statistical methods employed and specify the thresholds used. This information would greatly assist in interpreting the significance and reliability of the data.
General comments:
Since the study is exclusively “in silico”, it would be important to highlight the need to validate the physicochemical characteristics of the primary structure to trace a pattern of these results, in addition to verifying whether there are domains of other proteins present in these sequences. An analysis of conserved domains would help, as well as an analysis of secondary structure.
Among the more than 2,000 predicted peptides, what criteria were used to prioritize those with the greatest potential for further studies? Is there an analysis of the impact or functional relevance of these peptides?
It was not clear which tools were used in the prediction of activity. Was it done using which predictors? This needs to be clearly stated in the methodology.
The importance of the signal peptide in AMPs is well known. Was this analysis done? Together with or separately from the analysis of cysteines distribution?
The authors observed that α-hairpinins are widely distributed and share motifs with other AMPs. How did this impact the specificity of the predictions and the confidence in the results presented? The great diversity of motifs found may affect the specificity of the predictions and this may lead to some risk of including false positives in the list of putative α-hairpinins.
Regarding the discussion, the distribution of peptides among different plant families was discussed from a quantitative point of view. But how does this relate to the expected functionality of the peptides? Is there a correlation between the presence of α-hairpinins and specific biological characteristics of plants? For example, polyploids and diploids? Larger or smaller genomes? I believe this could be discussed, even if briefly.
The authors do not discuss or adequately highlight the limitations of their results, since they are characterized by the search for patterns (RegEx) followed by filters (using methods not reported in detail) to reach the candidate sequences for α-hairpinins.
I think the conclusions should highlight the difficulties of unequivocal identification of alpha hairpinins and highlight the need for careful curation after using the proposed tool, including steps that are not available in the presented tool.
Comments on the Quality of English LanguageThe english is sound.
Author Response
We would like to thank the reviewer for thorough reading of our manuscript and useful suggestions that led to significant improvement of the manuscript’s quality.
Reviewer 1.
The paper addresses one of the most challenging families of antimicrobial peptides, due to their complexity and structural variability. The authors have developed an interesting tool, but I would like them to clarify the following aspects:
- In page 10 (lines 405-406) the authors affirm: “The AMP families that could be revealed included defensins, thionins, cyclotides, snakins, hevein-like peptides and lipid-transfer proteins.”
My question is: if α-hairpinins share cysteine ​​motifs with other AMPs, this suggests that classification as α-hairpinins may be based on the presence of these structural motifs, and not necessarily on a unique function or a distinct category of peptide. It would be important for the authors to clarify how they clearly differentiate α-hairpinins from other peptides that share these motifs.
First of all, α-hairpinins have a specific Cys-motif, the most common one is CX{3}CX{4-20}CX{3}C, and a characteristic helix-loop-helix 3D structure is forming due to the presence of this motif. The main difference of this motif from other plant AMP Cys-motifs is that the distance between the first and the second, as well as the third and the fourth cysteines, is equal to three amino acid residues. The loop between the second and the third cysteines can be various. The described motif can be a part of a larger Cys-motif belonging to other AMP families, but in this case, the larger motif should have additional cysteines located not in such specific manner. That is, the motifs determining other AMP families will have additional cysteines in their full structure beyond CX{3}CX{4-20}CX{3}C, for example, CX{3}CX{4-20}CX{3}CX{4}CX{11}CX{3}C, but α-hairpinins should not include such additional cysteines.
However, much bigger problem for α-hairpinin prediction is that the precursors of these AMPs can contain several Cys-motifs, which can be potentially processed to correspondent number of mature α-hairpinins. In this case, algorithm has to search one Cys-motif, and then scan for the second α-hairpinin motif; it should allow additional single cysteines to be present out of α-hairpinin motifs. However, all these motifs will have the same general structure. Note, in Table 1 we added detailed inclusion and exclusion criteria that we used for identification of α-hairpinins in plant transcriptomes.
- In page 10 (lines 415-417), where the authors say: “For this reason, all the predicted peptides were additionally manually curated and annotated by BLAST search, and only the sequences that have passed all filtration steps were characterized as putative α-hairpinins.”
My concern is that the authors does not clearly detail the filtering steps after the BLAST search. To ensure the reproducibility and validity of their results, it is essential that the authors describe in detail all the criteria and methods used in filtering the sequences, including how the inclusion or exclusion of certain sequences was decided.
We added the description of filtering steps to the Methods section (mainly, Table 1). We also extended the pipeline algorithm description to facilitate the understanding of the filtration procedures used.
- In page 10 (lines 418-420), where the authors affirm: “We used the same set of 1,267 plant transcriptomes as in our previous investigations [29] in order to make the results of two studies comparable with each other and exclude non-hairpinin motifs as described above.”
Again, if α-hairpin motifs appear in other proteins, such as defensins, the paper needs to clarify whether these peptides were considered as α-hairpins or excluded, especially considering that defensins (for instance) may not be detected y BLAST. The methodology should include a clear explanation of how they dealt with these overlaps to avoid ambiguity.
We described the inclusion and exclusion criteria in Table 1. In general, the procedure is as follows.
|
Inclusion criteria |
Exclusion criteria |
|
1. Presence of 4C motif (CX{3}CX{4-20}CX{3}C) |
1. Absence of 4C/6C core motif. |
|
2. Presence of 6C motif CX{3}CX{3}CX{4-20}CX{3}CX{3}C |
2. Presence of CC before or after 4C/6C core motif. |
|
3. Presence of one or two additional cysteines, that can be located before or after 4C/6C core motif |
3. Presence before or after the core motif three and more cysteines which are not arranged in 4C/6C core motif. |
|
|
4. Presence X{1}CX{3}C after the motif. |
|
|
5. BLAST annotation as other protein, except vicilins and unknown proteins. |
Inclusion criteria were the presence of 4C or 6C motifs. The possibility of a presence of one or two additional cysteines before or after the motif was based on the reports that some precursors of α-hairpinins contained additional cysteines, which can be located either quite close to the source 4C/6C motif or in a significant distance from it.
Exclusion criteria were the absence of 4C/6C motifs. Furthermore, the presence of additional cysteine residues beyond α-hairpinin motif was also assessed. If double cysteines before or after the motif were present, this sequence was excluded, since this feature was a characteristic of other plant AMPs (thionins, hevein-like peptides, lipid-transfer proteins etc.). Besides, if there were three or more cysteines outside the core motifs, they had to be arranged in additional α-hairpinin motif or its part, otherwise this sequence was also excluded. The reason for such exclusion is that there are some plant AMP families, which have 4C/6C core motifs as a part of more broad Cys-motifs, but other cysteines in such motifs have a different arrangement. Another exclusion criterion was the presence of X{1}CX{3}C immediately after the motif, since this is the characteristic C-tail sequence of defensins. At last, if BLAST annotation characterized a sequence as some other non-hairpinin protein (e.g., thionin) except vicilins and unknown proteins, it was also excluded from further study.
- In page 10 (lines 431-432), Where the authors state: “Additional set of dedicated software (scripts) developed by us was used to collect the statistics on plant families possessing each type of motif.”
Where do the authors show these “additional scripts” or explain their specific functionalities? For a complete evaluation of the method, it would be necessary for the authors to detail what these scripts do, what kind of statistics are collected (such as motif frequency, distribution among species, etc.), as well as how this information was used in the analysis.
The scripts collect the motif distribution among plant families, species and parts. It takes the output file of the main pipeline and annotation file from 1kP project (1kP-Sample-List_2019-03.txt, also available on Github) as input and distributes the sequences revealed into groups based on the annotation provided. First level includes high-order groups (e.g. Core Eudicots), second level is for families, third level is for species, and the fourth is for tissues. The output is provided as text files in tabular form. These data is purely descriptive and does not involve analysis of statistically significant differences, so we changed the phrase to ‘collect descriptive statistics’. The script was uploaded to Github (cysmotif_stat.sh).
- In page 11 (lines 439-442): “Previously a sequence including additional cysteines beyond the motif was classified as belonging to artificial ‘cysteine-rich peptide’ class and excluded from further analysis due to uncertainty regarding its tertiary structure [29].”
I think that including a conserved domain evaluation could enhance the robustness of the findings. Given the brevity of the motifs, such an analysis would help mitigate potential mistranslation biases and ensure the validity of the conclusions. Could the authors elaborate on how they addressed this potential bias in the study?
We tried to reveal the conserved domains in alpha-hairpinin peptides. Previously, such domains were reported in Pfam. First example is Vicilin_N, pfam04702, the only member of the superfamily cl23732. Its description is as follows: «In Macadamia integrifolia, this region is processed into peptides of approximately 50 amino acids containing a C-X-X-X-C-(10-12)X-C-X-X-X-C motif. These peptides exhibit antimicrobial activity in vitro».
The second example is pfam14861: «Antimicrobial21 revealed in maize and barnyard grass and supposed to have an “alpha-helical hairpin fold stabilized by two disulphide bonds».
We tried to search for these two domains in our peptides, but found only 28 sequences with modest similarity in 4C peptides and 73 sequences within 6C peptides. Such a low number can be a result of low sequence similarity for the amino acids between cysteines. In addition, the uncertainty of proteolysis site location can contribute to hindrance in revealing such domains by homology search. As we stated in the manuscript, the lack of similarity is a greatest problem for revealing AMPs based on sequence homology only. We can suppose that if some domains were in fact the signatures of alpha-hairpinins, their sequences would have been revealed earlier by simple BLAST search.
- Page 11 (lines 445-446): “Then we collected the statistics on the motif distribution among plant families and parts of the plants. Full list of the peptide sequences revealed is available in Tables S1 (multidomain) and S2 (4C and 6C motifs).”
To facilitate a clearer understanding of the results, it would be beneficial to detail the statistical methods employed and specify the thresholds used. This information would greatly assist in interpreting the significance and reliability of the data.
We added the description to Methods section. Basically, we used chi-square test to check whether the alpha-hairpinin distribution depends on group factor, that is, plant family or organs. However, the only conclusion based in this data was a confirmation of Papaveraceae as a source of α-hairpinins.
General comments:
- Since the study is exclusively “in silico”, it would be important to highlight the need to validate the physicochemical characteristics of the primary structure to trace a pattern of these results, in addition to verifying whether there are domains of other proteins present in these sequences. An analysis of conserved domains would help, as well as an analysis of secondary structure.
The domain analysis was described in the answer to point 5. Unfortunately, it did not allow to verify the alpha-hairpinins.
- Among the more than 2,000 predicted peptides, what criteria were used to prioritize those with the greatest potential for further studies? Is there an analysis of the impact or functional relevance of these peptides?
We performed the prediction of possible antifungal and antibacterial activity for potential α-hairpinins using third-party software. Please find the details in the answer to point 9.
We believe that the predicted α-hairpinins likely possessing both antibacterial and antifungal activity have the greatest potential for further research; in Supplementary table 2 they are marked in yellow. In addition, all multidomain precursors that have similarity with vicilin or known α-hairpinins are also interesting objects for future studies (Supplementary Table 1).
- It was not clear which tools were used in the prediction of activity. Was it done using which predictors? This needs to be clearly stated in the methodology.
We used several third-party programs and servers to predict the possible activity of the peptides revealed. We added the data to table S2, and also added the description to Methods section. However, the reliable in silico prediction of antimicrobial/antifungal activity is a very complex task, and, to the best of our knowledge, currently no generally accepted activity prediction algorithm exists. Various machine-learning approaches seem to be perspective, but currently they did not outperform simple classifiers based on supervised training.
We evaluated antimicrobial activity of all predicted peptides using different tools, specifically antibacterial activity was assessed by AntiBP3 (https://doi.org/10.3390/antibiotics13020168, https://webs.iiitd.edu.in/raghava/antibp3 ), antifungal - by Antifp (https://doi.org/10.3389/fmicb.2018.00323, http://webs.iiitd.edu.in/raghava/antifp) and AfpTransferPred (https://doi.org/10.3390/ijms241210270, https://selectfight.org/afptransferpred/) . The results shown that 6C class had a greater fraction of peptides predicted to be antifungal in comparison to 4C (95% vs. 83%). In addition, the number of antifungal/antibacterial peptides can be higher than the predicted one due to uncertainty of proteolysis site location.
- The importance of the signal peptide in AMPs is well known. Was this analysis done? Together with or separately from the analysis of cysteines distribution?
Our pipeline includes checking for the presence of signal peptide using SignalP. This analysis is performed together with investigation of cysteine distribution, and the peptides not including signal peptide according to SignalP results are excluded from further processing. We added a more detailed description of this step to the Methods.
- The authors observed that α-hairpinins are widely distributed and share motifs with other AMPs. How did this impact the specificity of the predictions and the confidence in the results presented? The great diversity of motifs found may affect the specificity of the predictions and this may lead to some risk of including false positives in the list of putative α-hairpinins.
We added more detailed description for exclusion and inclusion criteria. In brief, other AMPs like defensins can share a part of alpha-hairpinin motif, but they have additional cysteines, by which such motifs can be clearly distinguished. Please see Table 1 in the manuscript and the answer to the point 3 above.
- Regarding the discussion, the distribution of peptides among different plant families was discussed from a quantitative point of view. But how does this relate to the expected functionality of the peptides? Is there a correlation between the presence of α-hairpinins and specific biological characteristics of plants? For example, polyploids and diploids? Larger or smaller genomes? I believe this could be discussed, even if briefly.
We have studied the possible correlation of plant characteristics with the number of α-hairpinins revealed in such plants, but, unfortunately, have not revealed any significant correlations. For example, some diploids included lower number of α-hairpinins than the polyploids.
Papaveraceae family was the most abundant with α-hairpinins. In particular, Papaver setigerum and P. somniferum contained 31 and 43 predicted α-hairpinins with 4C motif and 6C motifs, respectively. However, these species differ in genome size: P. setigerum has almost 4.9 Gb and P. somniferum — only 2.7 Gb, and it is thought that P. setigerum is tetraploid (2n = 44) , while P. somniferum is the diploid (2n = 22) (10.1038/s41598-021-04056-3). Therefore, we cannot make a conclusion that the bigger is the genome, the higher is the number of α-hairpinins in it. As to the plant families, the distribution of 4C predicted α-hairpinins is shown in in Fig.1B. It is easy to see that Papaveraceae family was the most abundant with 4C α-hairpinins (the example of genome sizes is listed above); Chlamydomonadaceae was in the second place with a genome sizes around 120 Mb (10.1126/science.1143609). Moreover, almost the same number of predicted peptides was found in Poaceae, which genome sizes are considerably larger. Thus, we decided not to discuss any possible correlations in this sense.
On the other hand, we find some unusual distribution properties for the plant families and added the notion of this to the manuscript (lines 317-327). However, the only interesting point was the confirmation of Papaveraceae as a source of α-hairpinins.
- The authors do not discuss or adequately highlight the limitations of their results, since they are characterized by the search for patterns (RegEx) followed by filters (using methods not reported in detail) to reach the candidate sequences for α-hairpinins.
We added more detailed description of the methods used. They were described previously in the paper concerning Cysmotif searcher (10.1134/S0006297918110135), but we agree that understanding the process of filtering is a key point to assess the reliability of our algorithm. Besides searching for patterns, the algorithm also includes checking for the presence of correct ORF, the presence of signal peptide (signalP) and filtering the length of a mature peptide. We also added the discussion of the method and results limitation to Discussion section.
- I think the conclusions should highlight the difficulties of unequivocal identification of alpha hairpinins and highlight the need for careful curation after using the proposed tool, including steps that are not available in the presented tool.
Thank you for the suggestion. We added these statements to conclusion, and they in fact will help readers to better understand what to expect from the results of our algorithm and how to use them in their research.
Reviewer 2 Report
Comments and Suggestions for Authors
Overall, this work is written in a review format; the algorithm used for their predictions is unclear, and I suggest a better description of the alpha-hairpinins. Some structure, rather than 2D diagrams, and some description of the mechanisms of action of these peptides could be included. The presence of cysteines seems to be a determining factor and this issue is not addressed at all. I do not recommend publication in its current form.
Comments on the Quality of English LanguageAverage
Author Response
We would like to thank the reviewer for taking time to read our manuscript and the comments provided.
Reviewer 2.
Overall, this work is written in a review format; the algorithm used for their predictions is unclear, and I suggest a better description of the alpha-hairpinins. Some structure, rather than 2D diagrams, and some description of the mechanisms of action of these peptides could be included. The presence of cysteines seems to be a determining factor and this issue is not addressed at all. I do not recommend publication in its current form.
We added the extended description of alpha-hairpinin family and possible mechanisms of action for these peptides. Such mechanisms in most cases are hypotheses, which have not been verified yet.
We broadened the Methods section and described in detail how the algorithm worked. We also placed to Abstract, Introduction and Methods a thorough description of α-hairpinins. In the Methods section we payed specific attention to presence and location of cysteines in the primary structure of α-hairpinins.
In general, we presented a motif-based classifier, which allows to narrow the search for probable alpha-hairpinin sequences, but not to present a ready-to-use solution for determining the exact structure and function of a particular peptide based on its sequence only, which is hardly possible. We added a more detailed description of the algorithm used (Methods section).
We did not get the point why the respected reviewer considered our manuscript as a review, except for, possibly, large volumes of data and comprehensiveness of the analysis. All the data presented is novel and original, and alpha-hairpinins were predicted using novel motifs refined by us. To the best of our knowledge, currently no software or algorithm exist for specific prediction of alpha-hairpinins. In addition, two recent publications report the using the application of our algorithm to successful prediction of alpha-hairpinins, one of which includes in vitro antibacterial activity testing (https://doi.org/10.15406/ijmboa.2024.07.00179, https://doi.org/10.1021/acs.jproteome.3c00597).
The presence of cysteines is, without any doubt, the most significant and determining factor in ‘cysteine motifs’, and this issue was addressed throughout the manuscript starting from introduction and ending with a discussion. As for the role of cysteines in stabilizing the antimicrobial peptide structure, we added a brief statement (“The structure of many AMPs was previously shown to be stabilized by disulfide bonds, and cysteines play an essential role in such bond formation”), but this is a general knowledge in the field.
Reviewer 3 Report
Comments and Suggestions for Authors
Here are a few concerns on the manuscript by Slavokhotova et al. submitted to Antibiotics:
1. The abstract mentions “α-hairpinins” as a family of antimicrobial peptides but does not clearly define what α-hairpinins are.
2. The results section discusses different motifs (e.g., 4C, 4C+1, 4C+2, 5C, 6C) without fully explaining how these variations affect the antimicrobial activity of the peptides. It states that peptides with a 6C motif may be more potent than those with a 4C motif, but this assertion lacks sufficient evidence or explanation.
3. There is a lack of actual activity of these peptides in “in vivo” or “in vitro”.
4. The method mentions that the “Cysmotif searcher” pipeline was used to identify α-hairpinins, but it does not describe how the accuracy of this pipeline was validated for α-hairpinin prediction. The methodology lacks any performance metrics (e.g., sensitivity, specificity, precision) to show how well the pipeline performs in identifying true α-hairpinins versus false positives. This omission raises concerns about the reliability of the results.
5. The pipeline includes manual curation of the predicted sequences by BLAST search, but the criteria for this manual curation are not clearly defined. How were sequences evaluated during manual curation, and what specific thresholds were used to determine which sequences to keep or discard?
6. The methodology briefly mentions that motifs were “deduced from literature and AMP databases,” but it does not provide details on the process of motif deduction. How were these motifs selected, and what criteria were used to ensure they accurately represent α-hairpinins? This lack of detail makes it difficult to assess the robustness of the motif deduction process, potentially leading to incomplete or incorrect motif identification.
7. The study relies heavily on data from the 1KP project, using the same set of 1,267 plant transcriptomes as in previous investigations. However, it does not mention any efforts to update or expand this dataset with more recent or diverse transcriptomic data. By not incorporating additional data, the study may miss out on identifying new or diverse α-hairpinin sequences, limiting the generalizability of the findings.
8. It is mentioned that a “semi-automated algorithm” was used, but the methodology does not explain how this algorithm works or how it was tested. Without sufficient detail, it’s difficult to understand the effectiveness and reliability of this approach. Additionally, modular motifs with similar but not identical sequences could introduce false positives.
9. The methodology mentions the collection of statistics on motif distribution among plant families but does not describe how these statistics were calculated or analyzed. There is no mention of the statistical methods used to assess the significance of the findings, making it unclear whether the observed distributions are meaningful or due to chance.
10. The method states that motifs found in only one sequence were excluded to “increase the reliability of the analysis,” but this exclusion criterion might be too stringent. Unique sequences could represent rare but valid α-hairpinins, and excluding them could bias the analysis toward more common sequences, potentially missing novel discoveries. Additionally, the decision to exclude sequences based solely on frequency without other supporting evidence is not well-justified.
Comments on the Quality of English LanguageOverall, the English quality is good. Some commas, punctuation, and articles need to be included.
Author Response
We would like to thank the reviewer for thorough reading of our manuscript and useful suggestions that led to significant improvement of the manuscript’s quality.
Reviewer 3.
Here are a few concerns on the manuscript by Slavokhotova et al. submitted to Antibiotics:
- The abstract mentions “α-hairpinins” as a family of antimicrobial peptides but does not clearly define what α-hairpinins are.
In the Abstract line 11, we included, “α-Hairpinins are known as short peptides contained four cysteine residues that are arranged in the specific Cys-motif. These AMPs have a characteristic helix−loop−helix structure with two disulfide bonds”.
We also included a more detailed description of α-hairpinins in Introduction starting at line 54 up to the end of the paragraph. Currently the description is:
“The latter are small peptides having a very specific Cys-motif that can be denoted as C1X3C2XnC3X3C4, where Ci shows cysteine residues (i=1…4), X can independently be any amino acid residue except cysteine, while subscripts 3 and n indicate the number of such non-cysteine residues. This cysteine arrangement facilitates common helix−loop−helix spatial structure comprising two α-helices oriented antiparallel and joined by a loop []. Motif sequences can be highly heterogeneous even within the same plant species, with cysteines sometimes being the only conservative residues, which fact significantly complicates the possibility of detecting a new α-hairpinin using a homology search only. It should be noted that until recently α-hairpinins were considered to contain only four cysteines. However according to the primary Cys-motif of the precursors, the fifth or sixth cysteines located one to three amino acids apart from the main ones can also be presented. Moreover, novel antibacterial α-hairpinin with six cysteines has been recently discovered in a ghost pepper [https://doi.org/10.1021/acs.jproteome.3c00597)/]. In addition, α-hairpinin with six cysteines and characteristic helix−loop−helix structure was isolated from Nigella sativa (PDB ID 2NB2, https://www.rcsb.org/structure/2nb2). Thus we can conclude that α-hairpinins are peptides with specific 4-6 cysteine residue motif sharing common helix−loop−helix structure.”
- The results section discusses different motifs (e.g., 4C, 4C+1, 4C+2, 5C, 6C) without fully explaining how these variations affect the antimicrobial activity of the peptides. It states that peptides with a 6C motif may be more potent than those with a 4C motif, but this assertion lacks sufficient evidence or explanation.
We would like to pay the attention that this work was held fully in silico. We added the word “computational” to the title of the manuscript to not mislead the readers. Furthermore, we evaluated antimicrobial activity of all predicted peptides using different tools, specifically antibacterial activity was assessed by AntiBP3 (https://doi.org/10.3390/antibiotics13020168, https://webs.iiitd.edu.in/raghava/antibp3 ), antifungal - by Antifp (https://doi.org/10.3389/fmicb.2018.00323, http://webs.iiitd.edu.in/raghava/antifp) and AfpTransferPred (https://doi.org/10.3390/ijms241210270, https://selectfight.org/afptransferpred/) . The results shown that 6C class had a greater fraction of peptides predicted to be antifungal in comparison to 4C (95% vs. 83%). We changed the Results section to reflect these findings.
- There is a lack of actual activity of these peptides in “in vivo” or “in vitro”.
As we stated in Abstract, currently there are just 12 α-hairpinin family representatives with activity proven in vitro or in vivo. Our analysis was performed only in silico to facilitate future activity determination experiments, but performing such experiments lies beyond the scope of the manuscript. In order to facilitate such future experiments and make results more reliable, we added the results of antifungal/antibacterial activity estimation performed by third-party classification software for the peptides revealed by us (see Table S2, worksheets ‘4C_predicted_activity’ and ‘6C_predicted_activity’), and the rate of the predicted antifungal activity was rather high for both groups (83% for 4C and 95% for 6C). We also added two recent references, in which other researchers revealed alpha-hairpins using software or motifs developed by us (https://doi.org/10.15406/ijmboa.2024.07.00179, https://doi.org/10.1021/acs.jproteome.3c00597.) In the latter work by Culver et al., the researchers discovered novel 6C peptide from ghost pepper using Cysmotif searcher prediction, tested it and its analogues and found that the peptides in fact inhibited ESKAPE pathogens.
- The method mentions that the “Cysmotif searcher” pipeline was used to identify α-hairpinins, but it does not describe how the accuracy of this pipeline was validated for α-hairpinin prediction. The methodology lacks any performance metrics (e.g., sensitivity, specificity, precision) to show how well the pipeline performs in identifying true α-hairpinins versus false positives. This omission raises concerns about the reliability of the results.
As we mentioned in the manuscript, the α-hairpinin family currently includes only 12 representatives with proven antimicrobial activity, although some researchers suppose that this number represents only a small fraction of real peptides. Thus, it is difficult, if not impossible, to make a reliable estimate of sensitivity/specificity of α-hairpinin prediction based on the already annotated data. Our pipeline reveals all previously known α-hairpinins with proven activity, but we cannot determine, which of the peptides predicted by us are false positives. In order to check the reliability of our predictions, we used several web-servers predicting antifungal/antimicrobial activity (see above). We also performed the prediction on randomly generated sequences with the same amino acid frequency characteristics, as in the peptides we revealed, to estimate the probability of revealing α-hairpinin by chance. We added all these data to the manuscript (Methods section, Table S2).
- The pipeline includes manual curation of the predicted sequences by BLAST search, but the criteria for this manual curation are not clearly defined. How were sequences evaluated during manual curation, and what specific thresholds were used to determine which sequences to keep or discard?
We added a Table 1 containing inclusion and exclusion criteria for a sequence to be referred as a α-hairpinin.
|
Inclusion criteria |
Exclusion criteria |
|
1. Presence of 4C motif (CX{3}CX{4-20}CX{3}C) |
1. Absence of 4C/6C core motif. |
|
2. Presence of 6C motif CX{3}CX{3}CX{4-20}CX{3}CX{3}C |
2. Presence of CC before or after 4C/6C core motif. |
|
3. Presence of one or two additional cysteines, that can be located before or after 4C/6C core motif |
3. Presence before or after the core motif three and more cysteines which are not arranged in 4C/6C core motif. |
|
|
4. Presence X{1}CX{3}C after the motif. |
|
|
5. BLAST annotation as other protein, except vicilins and unknown proteins. |
Inclusion criteria were the presence of 4C or 6C motifs. The possibility of a presence of one or two additional cysteines before or after the motif was based on the report that some precursors of α-hairpinins contained additional cysteines, which can be located either quite close to the source 4C/6C motif or in a significant distance from it.
Exclusion criteria were the absence of 4C/6C motifs. Furthermore, the presence of additional cysteine residues beyond α-hairpinin motif was also assessed. If double cysteines before or after the motif was present, this sequence was excluded, since this feature was a characteristic of other plant AMPs (thionins, hevein-like peptides, lipid-transfer proteins etc.). Besides, if there were three or more cysteines outside the core motifs, they had to be arranged in additional α-hairpinin motif or its part, otherwise this sequence was excluded. The reason of such exclusion is that there are some plant AMP families, which have 4C/6C core motifs as a part of more broad Cys-motifs, but other cysteines in such motifs located in different arrangement. Another exclusion criteria was the presence of X{1}CX{3}C immediately after the motif, since this is the characteristic C-tail sequence of defensins. At last, if BLAST annotation characterized a sequence as some other non-hairpinin protein (e.g., thionin) except vicilins and unknown proteins, it was also excluded from further study.
- The methodology briefly mentions that motifs were “deduced from literature and AMP databases,” but it does not provide details on the process of motif deduction. How were these motifs selected, and what criteria were used to ensure they accurately represent α-hairpinins? This lack of detail makes it difficult to assess the robustness of the motif deduction process, potentially leading to incomplete or incorrect motif identification.
The details were added in the answer to previous issues (inclusion/exclusion criteria, activity checking).
- The study relies heavily on data from the 1KP project, using the same set of 1,267 plant transcriptomes as in previous investigations. However, it does not mention any efforts to update or expand this dataset with more recent or diverse transcriptomic data. By not incorporating additional data, the study may miss out on identifying new or diverse α-hairpinin sequences, limiting the generalizability of the findings.
1kp project is a regularly updated database of high quality plant transcriptomes, and thus we believe it is suitable for searching of generalized patterns within amino acid sequences. Our research is focused only on plant antimicrobial peptides, and this project seems to have the most diverse collection of plants. In fact, we could miss out some highly diverse α-hairpinin sequences present in some new plant transcriptomes, but our goal was to capture the general patterns for this family, not to predict as much potentially active peptides as possible. Since reliable in silico prediction of antimicrobial/antifungal activity is a very complex task, we believe that provision of more reliable data will better facilitate future ‘wet lab’ activity checking experiments than outputting more rare, yet less reliable, motifs and/or peptides.
- It is mentioned that a “semi-automated algorithm” was used, but the methodology does not explain how this algorithm works or how it was tested. Without sufficient detail, it’s difficult to understand the effectiveness and reliability of this approach. Additionally, modular motifs with similar but not identical sequences could introduce false positives.
We added the description of the algorithm to the Methods section. It generally includes running cysmotif_searcher pipeline in several rounds, where the output of previous round goes to the input of the next. We also added scripting commands for this algorithm to main Github repository (cysmotif_modular.sh).
- The methodology mentions the collection of statistics on motif distribution among plant families but does not describe how these statistics were calculated or analyzed. There is no mention of the statistical methods used to assess the significance of the findings, making it unclear whether the observed distributions are meaningful or due to chance.
Previously we reported only descriptive statistics without performing significance estimation. In the updated manuscript, we added the chi-square test estimation for the cases where enough data can be obtained to check the motif distribution. However, the only conclusion based in this data was a confirmation of Papaveraceae as a source of α-hairpinins (see section 3.2 for details)
- The method states that motifs found in only one sequence were excluded to “increase the reliability of the analysis,” but this exclusion criterion might be too stringent. Unique sequences could represent rare but valid α-hairpinins, and excluding them could bias the analysis toward more common sequences, potentially missing novel discoveries. Additionally, the decision to exclude sequences based solely on frequency without other supporting evidence is not well-justified.
As we mentioned in the answer to previous questions above, our goal was to provide future activity determination experiments with more reliable and generalized data, and not to capture the whole possible diversity of potential α-hairpinins. Single sequences can represent rare peptides, but they can also arise from sequencing errors or artifacts introduced by transcriptome assembly software. However, we do understand that such sequences can be of interest for some researchers, so we added them to separate table in Supplementary data (table S3), but we did not include them in the downstream analysis since they are probably not reflecting general properties of the families or groups of plants to which they belong.
Round 2
Reviewer 1 Report
Comments and Suggestions for Authors
The article provides an interesting and necessary tool.
The revised version of the article added relevant information to the methodology, including previously missing descriptions and a supplementary table that now make it possible to better understand the filtering and elimination steps of other peptides with similar patterns.
I consider that the article can be accepted in its current version.
Author Response
No corrections were suggested.
Reviewer 2 Report
Comments and Suggestions for Authors
1. The introduction has improved concerning the previous version, however, I suggest including in the introduction section a first figure showing the 3D structure of some of these a-hairpinins, perhaps the one reported for Nigella, for instance.
2. Hand in hand with the previous point, I also suggest including a sequence logo of the a-hairpinins to make evident the conservation of the motifs 4C, 5C, 6C for example.
3. Improve the wording of the paragraph beginning on line 147 of this manuscript.
4. The algorithm used for the prediction of putative α-hairpinins is not very clear to this reviewer, this aspect of the algorithm description could be improved in the corresponding section. The use of a flowchart is suggested.
5. Given the great diversity of these peptides, the description of a representative peptide for each family, in terms of sequence/structure, is recommended. The excessive use of acronyms and numbers used in Figures 3 and 4 makes the reading of this text so difficult.
Comments on the Quality of English LanguageNo more comments
Author Response
Reviewer 2
- The introduction has improved concerning the previous version, however, I suggest including in the introduction section a first figure showing the 3D structure of some of these a-hairpinins, perhaps the one reported for Nigella, for instance.
We added Fig. 1 with the structures of EcAMP1 and Nigellin-1 to Introduction.
- Hand in hand with the previous point, I also suggest including a sequence logo of the a-hairpinins to make evident the conservation of the motifs 4C, 5C, 6C for example.
We have added the sequence logos for the most prevalent 4C and 6C motifs as Fig.2 and 4, respectively
- Improve the wording of the paragraph beginning on line 147 of this manuscript.
The paragraph was rewritten as suggested.
- The algorithm used for the prediction of putative α-hairpinins is not very clear to this reviewer, this aspect of the algorithm description could be improved in the corresponding section. The use of a flowchart is suggested.
Flowchart was added as Figure 6.
- Given the great diversity of these peptides, the description of a representative peptide for each family, in terms of sequence/structure, is recommended. The excessive use of acronyms and numbers used in Figures 3 and 4 makes the reading of this text so difficult.
We deleted Fig. 3 and replaced acronyms with the full name of plants and precursors in Fig. 4 (now this is the figure 5). We also deleted all acronyms in Results and Discussion to improve the readability.
Reviewer 3 Report
Comments and Suggestions for Authors
I think the authors almost resolved the concerns I raised.
Author Response
No corrections were suggested.